# Polaritonic molecular clock for all-optical ultrafast imaging of wavepacket dynamics without probe pulses

R.E.F. Silva [1]✉, Javier del Pino[2], Francisco J. García-Vidal[1,3] & Johannes Feist[1]✉

Conventional approaches to probing ultrafast molecular dynamics rely on the use of synchronized laser pulses with a well-defined time delay. Typically, a pump pulse excites a molecular wavepacket. A subsequent probe pulse can then dissociate or ionize the molecule, and measurement of the molecular fragments provides information about where the wavepacket was for each time delay. Here, we propose to exploit the ultrafast nuclear-position-dependent emission obtained due to large light–matter coupling in plasmonic nanocavities to image wavepacket dynamics using only a single pump pulse. We show that the time-resolved emission from the cavity provides information about when the wavepacket passes a given region in nuclear configuration space. This approach can image both cavity-modified dynamics on polaritonic (hybrid light–matter) potentials in the strong light–matter coupling regime and bare-molecule dynamics in the intermediate coupling regime of large Purcell enhancements, and provides a route towards ultrafast molecular spectroscopy with plasmonic nanocavities.

[1] Departamento de Física Teórica de la Materia Condensada and Condensed Matter Physics Center (IFIMAC), Universidad Autónoma de Madrid, E-28049 Madrid, Spain. [2] Center for Nanophotonics, AMOLF, Science Park 104, 1098 XG Amsterdam, The Netherlands. [3] Donostia International Physics Center (DIPC), E-20018 Donostia/San Sebastián, Spain. ✉email: ruiefdasilva@gmail.com; johannes.feist@uam.es

The interaction of light and matter is one of the most fundamental ways to unveil the laws of nature and also a very important tool in the control and manipulation of physical systems. When a confined light mode and a quantum emitter interact, the timescale for the energy exchange between both constituents can become faster than their decay or decoherence times, and the system enters the strong coupling regime[1–3]. In this regime, the excitations of the system become hybrid light–matter states, the so-called polaritons, separated by the vacuum Rabi splitting $\Omega_R$. Due to their relatively large dipole moments and large exciton-binding energies, strong coupling can be achieved with organic molecules at room temperature down to the few- or even single-molecule level[4–6]. Strong coupling can lead to significant changes in the behavior of the coupled system, affecting properties such as the optical response[3–12], energy transport[13–16], chemical reactivity[17–25], and intersystem crossing[22,26,27]. However, up to now these setups did not provide direct information on the molecular dynamics.

A well-known approach to directly probe molecular dynamics is through the use of ultrashort coherent laser pulses, pioneered in the fields of femtochemistry[28] and attosecond science[29]. This allows to observe and control nuclear and electronic dynamics in atoms and molecules at their natural timescale (fs and sub-fs), and is a fundamental tool towards a better understanding of chemical and electronic processes[28–34]. In particular, real-time imaging of molecular dynamics can be achieved in experiments with a pump–probe setup with femtosecond resolution combined with the measurement of photoelectron spectra[31]. Although similar approaches could, in principle, provide a dynamical picture of molecules under strong light–matter coupling[35–37], common molecular observables (such as dissociation or ionization yields, or photoelectron spectra) are difficult to access in typical experimental setups, with molecules embedded in a solid-state matrix and confined within nanoscale cavities[4–6]. Another powerful approach is given by transient absorption spectroscopy, where the change of the absorption spectrum of a probe pulse is monitored as a function of time delay after a pump pulse. Although this can provide significant insight about molecular dynamics[38], the interpretation of the spectra is nontrivial due to the competition between several distinct effects (such as ground-state bleach, stimulated emission, and excited-state absorption) in the spectrum[39], such that transient absorption spectroscopy only gives an indirect fingerprint of the molecular dynamics.

In this study, we demonstrate that the ultrafast emission induced by strong coupling to plasmonic modes can be used to monitor molecular wavepacket dynamics by measuring the time-resolved light emission of the system after excitation by an ultrashort laser pulse, without the need of a synchronized probe pulse. Our approach exploits the fact that the light–matter hybridization in a molecule is nuclear-position-dependent. Consequently, efficient emission only occurs in regions where the polaritonic potential energy surface (PoPES)[18,40] on which the nuclear wavepacket moves possesses a significant contribution of the cavity mode, as sketched in Fig. 1. In addition, due to the very low lifetime (or, equivalently, low quality factor) of typical plasmonic nanocavity modes on the order of femtoseconds, emission from the cavity also becomes an ultrafast process. Instead of using a probe pulse to learn where the nuclear wavepacket is at a given time delay, we thus use the nuclear-position-dependent emission to learn when the wavepacket passes a given spatial region. Tracking the time-dependent emission from the cavity then gives direct information about the nuclear dynamics by effectively clocking the time it takes the wavepacket to perform a roundtrip in the PoPES through an all-optical measurement. We note that a variety of experimental techniques allow the measurement of time-dependent light

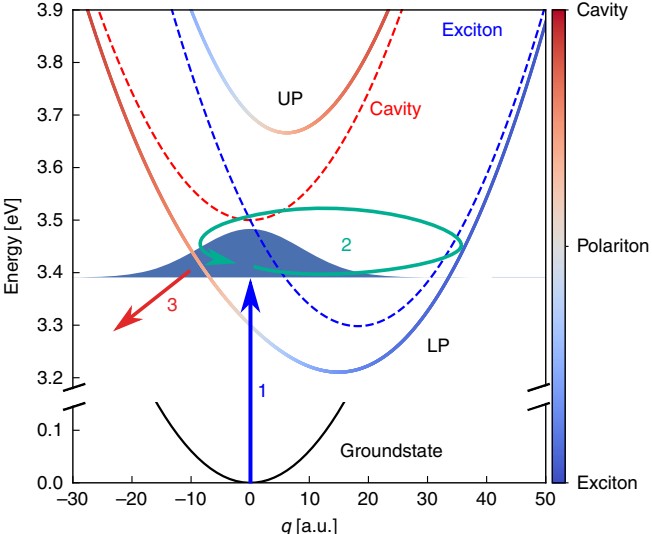

**Fig. 1 Polaritonic potential energy surface (PoPES).** PoPES in the single-excitation subspace for a single molecule coupled to a confined light mode. The red dashed line represents the uncoupled potential energy surfaces (PES) for a ground-state molecule with a single photon in the cavity, whereas the blue dashed line represents the molecular excited-state PES with no photons present and the black solid line represents the ground-state PES of the molecule. The solid blue/gray/red curves are the lower and upper polariton PES, with the color encoding the excitonic/polaritonic/photonic character as a function of nuclear position $q$. The filled blue curve represents the vibrational ground-state wavefunction of the electronic ground-state PES. The arrows represent the excitation by the laser pulse (1), oscillatory motion of the excited vibrational wavepacket (2) and radiative emission (3).

pulses with few-femtosecond resolution, e.g., intensity cross-correlation[41], SPIDER[42], FROG[43], or d-scan[44].

## Results

**Single molecule.** We first illustrate these ideas using a minimal model system: a single-mode nanocavity containing a molecule with two electronic states and a single vibrational degree of freedom, which for simplicity we approximate as a harmonic oscillator (with displacement between the ground and excited state due to exciton–phonon coupling). Our model is then equivalent to the Holstein–Jaynes–Cummings model that has been widely used in the literature to model strongly coupled organic molecules[19,45–47], with the main difference that we explicitly treat cavity losses and driving by an ultrashort (few-fs) laser pulse, and monitor the time-dependent emission. Although this is a strongly reduced model that allows for a straightforward interpretation, we will later show that the results we observe are also obtained in realistic simulations of molecules with a plethora of vibrational modes leading to rapid dephasing[10]. The system is described by the Hamiltonian (setting $\hbar = 1$)

$$H(t) = \omega_e \sigma^+ \sigma^- + \frac{p^2}{2} + \omega_v^2 \frac{q^2}{2} - \lambda_v \sqrt{2\omega_v} \sigma^+ \sigma^- q$$
$$+ \omega_c a^\dagger a + \frac{\Omega_R}{2}(a^\dagger \sigma^- + a \sigma^+) + \mu_c E(t)(a^\dagger + a), \quad (1)$$

where $\sigma^+$ ($\sigma^-$) is the raising (lowering) operator for the electronic state with excitation energy $\omega_e = 3.5$ eV, whereas $p$ and $q$ are the mass-weighted nuclear momentum and position operators for the vibrational mode with frequency $\omega_v = 0.182$ eV and exciton–phonon coupling strength $\lambda_v = 0.192$ eV (with these parameters, we reproduce the properties of the anthracene

molecule; see Methods for further details). The cavity is described through the photon annihilation (creation) operators $a$ ($a^\dagger$), with a photon energy chosen on resonance with the exciton, $\omega_c = \omega_e$. In addition to the coherent dynamics described by the Hamiltonian, the cavity mode decays with rate $\gamma_c = 0.1$ eV, described by a standard Lindblad decay operator (see Methods for details). The photon–exciton coupling is described through the Rabi splitting at resonance, $\Omega_R = 2E_{1ph}(r_m) \cdot \mu_{eg}$, where $E_{1ph}(r_m)$ is the quantized mode field of the cavity at the molecular position and $\mu_{eg}$ is the transition dipole moment of the molecule (in principle, this is $q$-dependent, but is taken constant here for simplicity). Finally, the cavity mode is coupled through its effective dipole moment $\mu_c$ to an external (classical) laser pulse $E(t) = E_0 \cos(\omega_L t) \exp(-\sigma_L^2 t^2/2)$, with central frequency $\omega_L$, spectral bandwidth $\sigma_L$, and a corresponding duration of $\approx 1.67/\sigma_L$ full width at half maximum (FWHM) of intensity. We note that as the cavity mode is driven by the external field, the effective pulse felt by the molecule (in particular in the weak coupling limit) is slightly distorted and not just given by $E(t)$.

We start by analyzing the system response in the strong coupling regime ($\Omega_R = 0.4$ eV) after excitation by an ultrashort laser pulse with $\sigma_L = 0.1$ eV, while scanning the laser frequency $\omega_L$. For $\sigma_L = 0.1$ eV, the duration of the pulse is $\approx 11$ fs. The laser intensity is chosen small enough to remain in the single-excitation subspace (i.e., within linear response). The instantaneous radiative emission rate from the cavity is given by $E_R = \gamma_{c,r}\langle a^\dagger a\rangle$, where $\gamma_{c,r}$ is the radiative decay rate of the cavity excitations. As it corresponds to a constant (system-dependent) factor, we set it to unity in the figures shown in the following. Estimates of the achievable photon yields in realistic systems are given in the Discussion section. In Fig. 2, the

time-dependent radiative emission intensity $E_R$ and the exciton population $\langle\sigma^+\sigma^-\rangle$ are shown. We observe that when the laser pulse is resonant with the lower polariton region, i.e., for $\omega_L$ between 3.2 and 3.5 eV, the cavity emission is modulated in time with a period of around 26 fs, whereas no such oscillation is observed when the upper polariton branch is excited for $\omega_L$ between 3.5 and 3.8 eV. This behavior can be understood with the help of the PoPES, shown in Fig. 1. They are obtained by treating nuclear motion within the Born–Oppenheimer approximation, i.e., with $q$ treated as an adiabatic parameter (see Methods for details). Within the Franck–Condon approximation, short-pulse excitation creates a copy of the vibrational ground state (centered at $q = 0$) on the relevant polaritonic PES. This vibrational wavepacket will then evolve on the potential surface, performing oscillatory motion, with the character of the wavepacket also oscillating between photon-dominated and exciton-dominated depending on nuclear position. However, as radiative emission of the cavity mode is orders of magnitude faster than from the bare molecule (typically, femtoseconds compared with nanoseconds), efficient emission is only possible in regions where the relevant PoPES has a significant photon contribution. Focusing first on the lower polariton, this condition is fulfilled for $q < 0$ for the parameters chosen here, explaining the observed temporal modulation of the emission intensity, which effectively corresponds to clocking of the nuclear wavepacket motion. Furthermore, the period of this motion is determined by the curvature of the lower polariton PoPES, which is different to the bare-molecule oscillation period $T_v \approx 22.7$ fs. Fitting the lower polariton curve to a harmonic oscillator for the current parameters gives an oscillation period of 25.9 fs, in excellent agreement with the observed modulation frequency of 26 fs. The temporal emission modulation thus also provides a direct fingerprint of the strong coupling-induced modifications of molecular structure. On the other hand, excitation to the upper polariton creates a wavepacket that spends most of its time in the region with efficient emission ($q > 0$ for the upper PoPES), such that no clear oscillation between photonic and excitonic character, and thus no modulation in the emission intensity, are observed.

Up to now, we have confirmed that molecular dynamics imprints its fingerprint in the time-dependent radiative emission of the cavity. We now demonstrate that the time-resolved emission intensity indeed provides a direct quantitative probe of the nuclear wavepacket dynamics. In Fig. 3, we show the nuclear probability density $|\psi(q)|^2$ in the single-excitation subspace under resonant excitation of the lower polariton, Fig. 3a, and upper polariton, Fig. 3b, respectively. For case (a), the wavepacket starts periodic motion around the minimum of the lower polariton curve, $q_{min} \approx 15$ a.u., after the initial excitation at $t \approx 0$. In the upper panel, we show $E_R$ and the probability to find the nuclei at $q \leq 0$, given by $\int_{-\infty}^0 |\psi(q)|^2 \mathrm{d}q$. The observed good agreement demonstrates that it is possible to track the position of the nuclear wavepacket in time through the emission from the cavity. The similarly good agreement found in Fig. 3b, with the integral this case performed for $q \geq 0$ corresponding to excitation of the upper polariton branch reinforces this notion. We again observe that a less-pronounced oscillation is observed for excitation of the UP branch. We also note that for case (b), there is a small contribution of the lower polariton to the excitation (as this is energetically still allowed), explaining the slightly worse agreement between the full calculation and the simplified approximation based on integrating the nuclear probability density.

We next investigate the dependence of the effects discussed above on the Rabi splitting $\Omega_R$, focusing in particular on the case of smaller $\Omega_R$, which would correspond to the weak coupling regime. The corresponding time-resolved radiative emission $E_R$ is shown in Fig. 4a on a logarithmic scale. As we have observed the

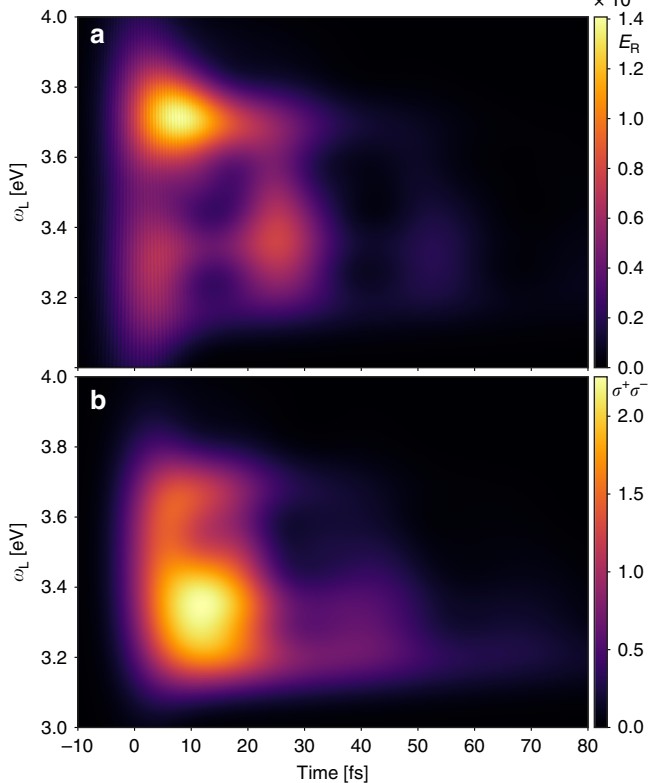

**Fig. 2 Time-dependent radiative emission.** Mean values of the time-dependent radiative emission, $E_R$, (**a**) and $\langle\sigma^+\sigma^-\rangle$ (**b**) for different values of $\omega_L$ and for $\Omega_R = 0.4$ eV. For all calculations, $E_0 = 2.1 \times 10^{-7}$ a.u. and $\sigma_L = 0.1$ eV.

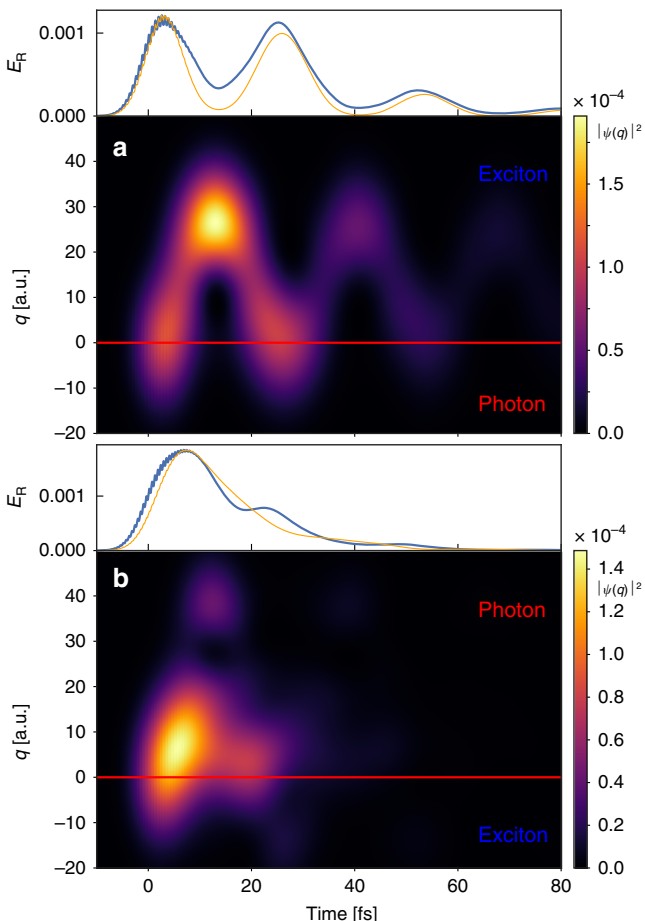

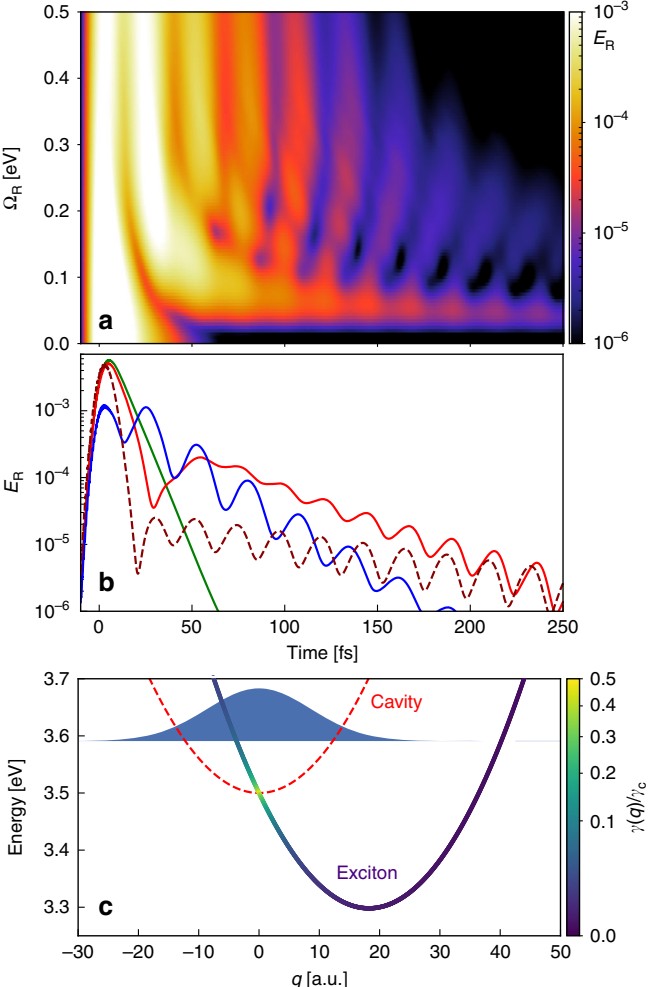

**Fig. 3 Tracking nuclear dynamics using the time-dependent radiative emission.** Probability density of the vibrational wavepackets in the single-excitation subspace for two different laser frequencies: (**a**) $\omega_L = 3.3$ eV and (**b**) $\omega_L = 3.7$ eV. The red line at $q = 0$ indicates the border where the polariton switches from photonic to excitonic character for the (**a**) lower and (**b**) upper polariton. The upper panels in each subfigure show the time-dependent emission from the cavity (thick blue lines) and the (scaled) probability of the nuclear wavepacket on the photonic side, given by $q < 0$ for **a** and $q > 0$ for **b** (orange lines). For both calculations, $E_0 = 2.1 \times 10^{-7}$ a.u. and $\sigma_L = 0.15$ eV.

**Fig. 4 From strong to weak coupling. a** Time-dependent radiative emission from the cavity, $E_R$, for different values of $\Omega_R$ and laser frequency resonant with the bare lower polariton energy, $\omega_L = \omega_e - \Omega_R/2$. For all calculations, $E_0 = 2.1 \times 10^{-7}$ a.u. and $\sigma_L = 0.15$ eV. **b** Same as in **a** for three different values of $\Omega_R = 0.01$, 0.07, and 0.4 eV (green, red, and blue lines, respectively). The dashed dark red line represents the case for $\Omega_R = 0.07$ eV and a cavity with larger decay rate, $\gamma_c = 0.3$ eV. **c** Potential energy surfaces in the weak coupling regime with large Purcell enhancement of the emission. The red dashed line represents the PES of the molecule in its ground state with a photon in the cavity. The blue–yellow solid line represents the molecular excited-state PES with no photons present, with the position-dependent (Purcell-enhanced) decay rate encoded in the purple/yellow color scale. The filled blue curve represents the vibrational ground-state wavefunction of the electronic ground-state PES.

lower polariton branch to display more interesting dynamics, the central laser frequency is chosen such that the lower polariton branch is excited for each Rabi frequency, i.e., $\omega_L = \omega_e - \Omega_R/2$. Several regimes can be clearly distinguished: for small coupling, $\Omega_R \lesssim 0.03$ eV, the molecules barely participate in the dynamics and the response is dominated by the excitation and subsequent ringdown (with time constant $\tau_c = \hbar/\gamma_c \approx 6.6$ fs) of the bare cavity mode (green line in Fig. 4b). In contrast, within the strong coupling regime, $\Omega_R \gtrsim 0.10$ eV, the previously discussed oscillations can be seen, with the modulation frequency increasing concomitantly with $\Omega_R$ due to the increasingly large modification of the polaritonic PES, and thus the nuclear oscillation period (blue line in Fig. 4b). For intermediate values of $\Omega_R$, a slightly different behavior is observed: emission occurs over relatively long times, but is again modulated over time, with a period of around 23 fs, in good agreement with the bare-molecule vibrational period $T_v \approx 22.7$ fs. This can be understood by examining the molecular PES in the case of weak coupling, as shown in Fig. 4c. In that case, the potential energy surfaces are almost unmodified and the initial laser pulse only excites the cavity mode, but the relatively large coupling is sufficient to

allow efficient energy transfer to the molecule (exactly in the Franck–Condon region) within the lifetime of the cavity mode, such that the emission is not fully dominated by the cavity response. The molecular wavepacket then again oscillates, now within the bare molecular excited-state PES. However, for nuclear configurations where the molecular exciton and the cavity mode are resonant (within the cavity bandwidth), the molecular radiative decay is enhanced strongly through the Purcell effect, leading to ultrafast emission exactly when the nuclear wavepacket crosses the resonant configuration ($q \approx 0$ for the parameters considered here). In the intermediate coupling regime, it is important to point out that the oscillations will be more clear when the cavity has an ultrafast decay. This can be seen when comparing the radiative emission for two different decay rates,

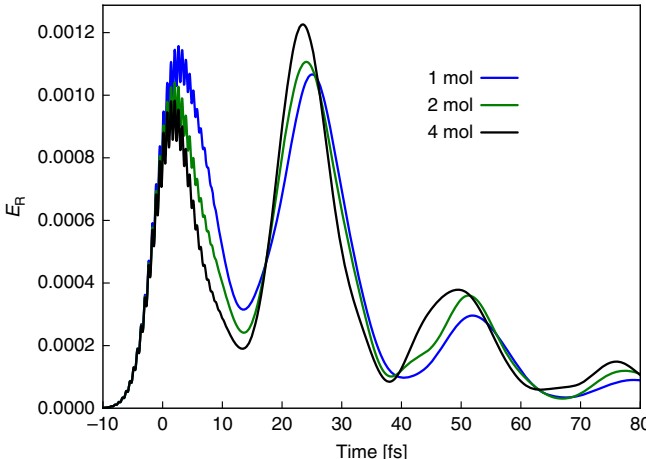

**Fig. 5 Multiple molecules.** Comparison of the radiative emission for different numbers of molecules coherently coupled to the same cavity mode while keeping the total Rabi splitting $\Omega_R \propto \sqrt{N}$ fixed. The blue, green, and black lines correspond to 1, 2, and 4 molecules, respectively. The frequency of the laser pulse is $\omega_L = 3.3$ eV and its bandwidth is $\sigma_L = 0.15$ eV.

$\gamma_c = 0.1$ and $0.3$ eV (solid red and dashed dark red lines in Fig. 4b), where the oscillations are more prominent for more lossy cavities. We note that the more relaxed requirements for $\Omega_R$ in this intermediate regime should make it more easily accessible in controlled experimental setups[6].

**Multiple molecules.** Up to now, we have focused on the case of a single molecule under strong coupling. Although this serves to highlight the principal properties of the setup, it is still extremely challenging to achieve in experiment. On the other hand, collective strong coupling can yield significant Rabi splittings in available plasmonic nanocavities even for small numbers of molecules (e.g., 200 meV for three or four molecules[5]). In this situation, several molecules are coherently coupled to the same photonic mode, with the collective Rabi splitting scaling as $\sqrt{N}$. In Fig. 5, we demonstrate that the polaritonic molecular clock also works in this situation. We plot the time-resolved radiative emission for $N = 1$, 2, and 4 molecules while keeping the collective Rabi splitting fixed at $\Omega_R = 0.4$ eV for easier comparison. This shows that the coherent wavepacket motion of multiple molecules moving on a collective PoPES can be accessed directly with our setup. We note that this is in strong contrast to standard pump–probe techniques, where only single-molecule observables are typically interrogated. In contrast, the PoPES in the case of collective strong coupling describe nuclear motion of the polaritonic supermolecule[48,49] and depend on all molecular coordinates. In the Supplementary Note 2, we use the time-dependent variational matrix product state (TDVMPS) approach[10,50] to show that this approach works even when taking into account all vibrational degrees of freedom and the associated dephasing. In particular, the effect of dephasing is not significantly stronger in the many-molecule case than for a single molecule. Consequently, the proposed setup could provide a route to directly probe multi-molecule coherent nuclear wavepacket motion.

**Non-harmonic potentials.** We next demonstrate that our approach is not restricted to displaced-harmonic oscillator models and also gives direct insight into molecular dynamics in more complex potentials. To that end, we treat a molecule described by displaced Morse potentials, corresponding to

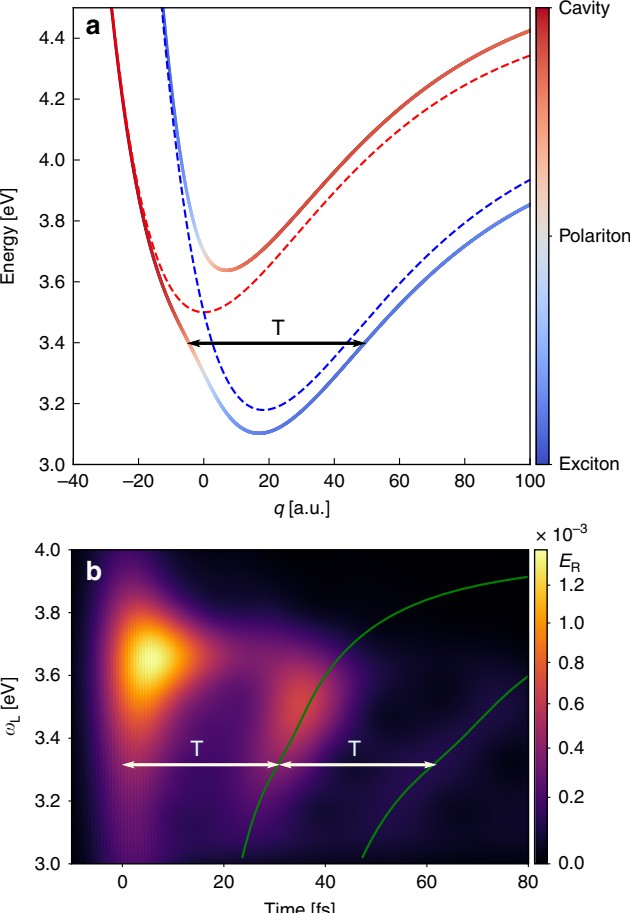

**Fig. 6 Morse potential. a** PoPES for a molecule where the ground and excited state are described by displaced Morse potentials. The colors and line styles have the same meaning as in Fig. 1. **b** Time-dependent radiative emission $E_R$ for different values of $\omega_L$ and for $\Omega_R = 0.4$ eV. For all calculations, $E_0 = 2.1 \times 10^{-7}$ a.u. and $\sigma_L = 0.1$ eV. The three green lines are calculated as $t = nT(\omega_L)$, with $n = 0$, 1, 2 (see main text).

anharmonic oscillators. The molecular Hamiltonian is given by

$$H_m = \frac{p^2}{2} + V_g(q)\sigma^- \sigma^+ + V_e(q)\sigma^+ \sigma^-, \tag{2}$$

$$V_j(q) = \delta_j + D\left(1 - e^{-a(q-q_j)}\right)^2, \tag{3}$$

where $j \in \{g, e\}$, with parameters $D = 1.0$ eV, $a = 0.025$ a.u., $q_g = 0$, $q_e = 18.2$ a.u., $\delta_g = 0$, and $\delta_e = 3.18$ eV. Figure 6a shows the uncoupled PES and the corresponding PoPES under strong coupling, whereas Fig. 6b shows the time-dependent radiative emission as a function of the driving laser frequency. In contrast with the simple displaced-harmonic-oscillator model treated before, the oscillation period of the time-dependent radiative emission now depends on the laser frequency. The insight this provides into the polaritonic PES becomes clear by comparing the peak times of the cavity emission with the energy-dependent classical oscillation period within the lower PoPES, $T(\omega_L) = 2\int_{q_{min}}^{q_{max}} dq \left[\frac{2}{M}\left(E_{gs} + \omega_L - V_{LP}(q)\right)\right]^{-1/2}$, where $E_{gs}$ is the ground-state energy and $M$ is the reduced mass. The green lines in Fig. 6b show that the peak emission happens exactly at $t = nT(\omega_L)$, with $n = 0$, 1, 2, ..., demonstrating that the polaritonic molecular clock captures the nuclear wave-packet motion accurately and provides a direct picture of the dynamics also in non-harmonic potentials. In the Supplementary Note 1, we furthermore show that our

scheme could also be used to study photodissocation dynamics within the weak coupling regime for a model molecule similar to methyl iodide[33].

We next discuss the requirements that must be fulfilled for the phenomena described above to be observed. First, the molecule needs to have sufficiently strong exciton–phonon coupling (i.e., a sufficiently large change in the $q$-dependent excitation frequency) to lead to significant spatial modulation of the cavity and exciton components of the PoPES. Furthermore, the slope of the (polaritonic) PES in the Franck–Condon region has to be large enough for the nuclear wavepacket to leave the initial position before it has time to decay completely (although this problem could be mitigated by, e.g., choosing the cavity to be resonant in another region of nuclear configuration space instead of at the equilibrium configuration). For the Holstein-type molecular model studied here, these conditions are satisfied if $\lambda_v$ is comparable to the vibrational frequency $\omega_v$, and both are comparable to the cavity decay rate $\gamma_c$. These properties are fulfilled for several organic molecules that have been used in strong coupling experiments, such as anthracene[51] or the rylene dye [N,N0-Bis(2,6-diisopropylphenyl)-1,7- and -1,6-bis (2,6-diisopropylphenoxy)-perylene-3,4:9,10-tetracarboximide][52]. In addition, to be able to observe coherent wavepacket motion, internal vibrational relaxation and dephasing, which typically occurs on the scale of tens to hundreds of femtoseconds in solid-state environments, must be slow enough compared with the dynamics of interest. In the Supplementary Note 2, we demonstrate that this is the case for the anthracene molecule by comparing the Holstein–Jaynes–Cummings model calculation with large-scale quantum dynamics simulations including all vibrational modes of the molecule, performed using the TDVMPS approach[10,50].

To summarize, we have proposed a scheme to probe and image molecular dynamics by measuring the time-dependent radiative emission obtained after short-pulse excitation of a system containing few molecules and a nanocavity with large light–matter coupling, close to or within the strong coupling regime. We show that this approach enables to retrieve a direct mapping of nuclear wavepacket motion in the time domain, also in the few-molecule case, where this scheme provides a direct fingerprint of coherent multi-molecular nuclear dynamics. In the strong coupling regime, this gives access to the cavity-modified molecular dynamics occurring on the PoPES, whereas in the weak coupling regime it allows probing of the bare-molecule excited-state dynamics. By exploiting the ultrafast emission dynamics in typical highly lossy plasmonic nanocavity, we obtain the time-resolved dynamics without the need for a pump–probe setup with synchronized femtosecond pulses. In addition, in contrast to the common approaches of femtochemistry, our proposed scheme does not require direct access to molecular observables such as photoelectron spectra or fragmentation yields, which are difficult to obtain for typical experimental geometries. Instead, it only relies on optical access to the nanocavity mode. In addition, the scheme only depends on the properties of the first few electronic states of the molecules, and is not affected by, e.g., the multitude of ionization channels that have to be taken into account in photoionization[34]. As only a single excitation is imparted to the molecules and the dynamics are probed through the photons emitted upon relaxation to the ground state, the molecules are left intact after the pulse. At the same time, this implies that the absolute photon numbers to be measured are small. This could be mitigated by using high-repetition-rate sources (readily available for the low laser intensities required), as well as collecting the response from an array of identical nanocavities, taking advantage of highly reproducible setups available nowadays, e.g., through DNA origami[6,53]. Finally, we mention that although the cavity decay rate $\gamma_c$ in a plasmonic cavity is typically large and leads to few-femtosecond lifetimes as required for the discussed approach,

this rate is often dominated by nonradiative contributions that do not lead to far-field emission. However, fortunately the same plasmonic nanocavity architectures that provide the current largest coupling strengths, such as nanoparticle-on-mirror geometries, also provide a significant radiative quantum yield of close to 50%[54–56].

## Methods

The time dynamics is described by the following Lindblad master equation

$$\dot{\rho}(t) = -i[H(t),\rho(t)] + \gamma_c \mathcal{L}_a[\rho(t)], \qquad (4)$$

where $\mathcal{L}_a[\rho(t)] = a\rho(t)a^\dagger - \frac{1}{2}[\rho(t)a^\dagger a + a^\dagger a\rho(t)]$ is a standard Lindblad decay term modeling the incoherent decay of the cavity mode due to material and radiative losses. The master equation numerical results were obtained using the QuTiP package[57,58]. The PoPESs used for the interpretation and analysis of the results are obtained by diagonalizing the (undriven) Hamiltonian within the Born–Oppenheimer approximation, i.e., diagonalizing $H(t) - p^2/2$ for $E_0 = 0$ and fixed $q$[18]. In Fig. 1, we show the PoPES within the single-excitation subspace, spanned by the uncoupled states $|e,0\rangle$ and $|g,1\rangle$, where $|g\rangle$ ($|e\rangle$) is the electronic ground (excited) state and $|n = 0,1,\ldots\rangle$ is the cavity mode Fock state with $n$ photons. The Hamiltonian in this subspace can be written as

$$H_{BO}(q) = \begin{pmatrix} \omega_c + \frac{\omega_v^2 q^2}{2} & \Omega_R/2 \\ \Omega_R/2 & \omega_e + \frac{\omega_v^2 q^2}{2} - \lambda_v\sqrt{2\omega_v}q \end{pmatrix}, \qquad (5)$$

and diagonalizing it gives the PoPES plotted in Figs. 1 and 4b.

The parameter values chosen for modeling the molecule were based on ab-initio calculations for the anthracene molecule at the TDA-B3LYP level of theory using Gaussian 09[59]. Fitting the PES obtained in these calculations to a displaced-harmonic oscillator model using the Duschinsky linear transformation[60],

$$H_{m,full} = \omega_e \sigma^+\sigma^- + \sum_k \left[\omega_k b_k^\dagger b_k + \lambda_k \sigma^+\sigma^-(b_k^\dagger + b_k)\right], \qquad (6)$$

yields the parameters $\{\omega_k, \lambda_k\}$, or equivalently the spectral density $J_v(\omega) = \sum_k \lambda_k^2 \delta(\omega - \omega_k)$, determining the vibrational spectrum of the molecule. The decoherence due to the coupling of the vibrational molecular modes with the surrounding bath is taken into account empirically by replacing the discrete peaks in the spectral density by a Lorentzian with 0.3 meV of width; however, the results are not affected by this. The single vibrational mode in Eq. (1) is then taken as the corresponding reaction coordinate, with $\lambda_v = \sqrt{\sum_k \lambda_k^2}$ and $\omega_v = \sum_k \omega_k \lambda_k^2/\lambda_v^2$[10,61].

In Fig. 7, we show the vibrational spectral density of anthracene (convoluted with a Lorentzian to represent broadening due to interactions with a solid-state environment). It can be seen that $\omega_v$ is very close to the frequency of the dominant vibrational mode in $J_v(\omega)$. We have additionally checked the validity of the single-mode approximation by comparing the model calculations above with TDVMPS) calculations[10,50] in which the full phononic spectral density, describing all vibrational modes of the molecule and surroundings, is taken into account (see Supplementary Note 1 for details).

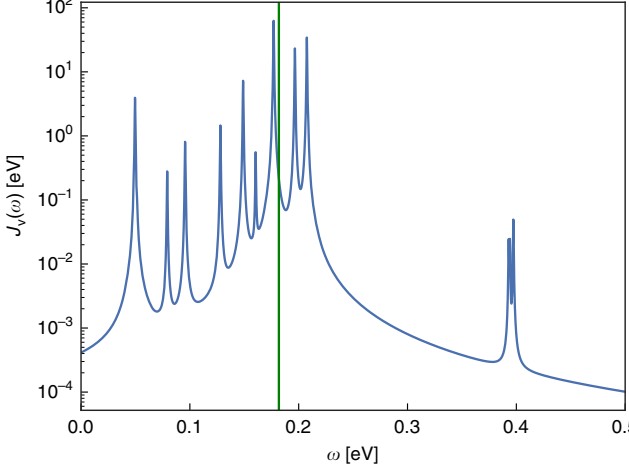

**Fig. 7 Anthracene spectral density.** $J_v(\omega)$ for the anthracene molecule. The vertical green line indicates the vibrational frequency $\omega_v$ of the reaction coordinate.

## Data availability

The data that support the plots within this paper and other findings of this study are available from the corresponding authors upon reasonable request.

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

## Acknowledgements

We thank Alex W. Chin and Florian A. Y. N. Schröder for their help with the TDVMPS calculations, and Clàudia Climent for her help with calculating the displaced-harmonic oscillator model and with the Duschinsky linear transformation. This work has been funded by the European Research Council grant ERC-2016-STG-714870 and the Spanish Ministry for Science, Innovation, and Universities—AEI grants RTI2018-099737-B-I00, PCI2018-093145 (through the QuantERA program of the European Commission), and CEX2018-000805-M (through the María de Maeztu program for Units of Excellence in R&D).

## Author contributions

R.E.F. S. and J.F. developed the idea. R.E.F.S. performed the numerical calculations. R.E.F.S., J.P., F.J.G.-V., and J.F. contributed to analysis of the results. R.E.F.S. and J.F. wrote the main part of the manuscript. The manuscript was discussed by R.E.F.S., J.P., F.J.G.-V., and J.F.

## Competing interests

The authors declare no competing interests.
