## [Peer Review File · Nature Communications]

Reviewers' Comments:

Reviewer #1:

Remarks to the Author:

The authors present an innovative idea to probe the motion of a nuclear wavepacket traveling on polaritonic states potential energy surfaces (PoPESs), without resorting to a probe pulse. Such effect is obtained by exploiting the enhanced spontaneous emission of a molecule coupled to a plasmonic nanocavity, provided that the resonant mode of the cavity is short-living. By detecting the emitted pulse, precious information on the excited state molecular processes can be accessed, even when the setup prevents to directly probe the sample, e.g. in molecules embedded in environments confined within nanoscale cavities.

The manuscript presents in detail the model, the system and the findings. In addition, the figures adequately support and illustrate the results. While the work is mainly polaritonic chemistry oriented, it also offers some interesting insight on the weak coupling case. Even more, it offers an original and useful way to exploit an experimental limit like short cavity lifetimes for useful applications. Even though the model is really simplified, it is tailored very conveniently to vehicle and thoroughly explain the original idea presented within this work.

I recommend the publication in Nature Communications after the following minor issues are taken care of:

1. The exact definition of the radiative emission E_R is not given. From the plot labels in the figures it appears to be adimensional, while I would have guessed it is the number of photon emitted per unit of time. Please clarify (in the main text).

2. There are two points that require a deeper discussion than what it is now in the manuscript:

2.a On p. 5 it is stated that the number of photon produced is low because a single excitation is imparted to the molecule. I think it should also be discussed that not all the photons lost from the cavity can really be detected, since a fraction of them will be irreversibly absorbed by the cavity material (this is certainly true for plasmonic nanocavities). Looking at the weak-coupling limit, the revealed emission from the nanocavity is essentially molecular fluorescence (although occurring well before full relaxation of the excited state), that may be quenched (i.e., most of the photons dissipated inside the cavity material) in this setup. The authors have certainly the tools to provide a quantitative estimate of this effect based on electromagnetic calculations on realistic plasmonic nanocavities, but a qualitative discussion of the point is also enough.

2.b Excitation of the molecule is described by a classical pulse whose central frequency is resonant with a hybrid light-molecule state, i.e., close to the frequency of the cavity mode. Wouldn't this deform the shape of the pulse? Isn't a description in which the external light source is actually injecting photons in the cavity a more faithful representation of the initial excitation conditions? Please comment

3 Details of the TDVMPs calculations should be provided in full to assure reproducibility (and understanding of what is really included), perhaps as SI. It would also be useful to have systematic insights in the interplay between coupling strengths and dephasing times on the expected signal.

4. The abstract makes reference to a class of pump-probe experiments (those where the probe dissociate or ionize the molecules) that is certainly important but not the only one, nor the one that "competes" more closely with the experiment suggested here. Pump-probe experiments where the probe pulse is a weak one, providing transient absorption data, is much more common for molecules

in condensed phases. I would rather (or additionally) compare the benefits of the present idea with this kind of experiments.

5. The PESs used here are harmonic. Nevertheless, a lot of interesting photochemical processes as photoisomerization or photoreactions would occur on PESs with more complex shapes. For the benefit of the impact of this work, a discussion of what one would expect to reveal for these more complex shapes would be useful, to avoid giving the impression the idea is limited to very simple PESs.

6. In Figure 3a there is a problem with the colorbar scale, as 5.7 and 5.2 appear between 0.0 and 1.0.

Reviewer #2:

Remarks to the Author:

In this manuscript the authors suggest a method which would allow one to map the nuclear wavepacket dynamics of a single molecule embedded in a nanocavity (for reaching or at least approaching strong coupling) by time-resolved monitoring of its emission after quasi-instantaneous excitation. Typically, pump-probe methods are used for studying this dynamics, the advantage of the suggested approach is that the optical scheme is simplified as only one optical pulse is needed. Moreover, the molecules would stay intact, to mention another advantage.

This is certainly an interesting concept, even though I am not sure that it is important enough to warrant publication in Nature Communications. Irrespective of this question I have a couple of points which are not clear to me:

1) The time scales are a bit hazy. In order to observe the calculated oscillations clearly, one needs a time-resolution of a fs or even better. This corresponds to a spectral width above 1 eV. On the other hand, for clarity one would like to monitor, for example, only the lower polariton, in particular as the presence of a broad vibrational mode spectrum blurs the oscillatory decay. But how is this possible, as the Rabi-splitting is well below (see also point 2) the spectral window that has to be monitored for obtaining the proper time-resolution? When selecting only the LPB, on the other hand, the time resolution would be typically above 10 fs, smearing out the described dynamics. Furthermore, doesn't the excitation laser pulse duration, assumed to be a few fs, wash out some of the effects? If using a 1 fs-pulse (hard to achieve at 3.5 eV), one would excite both LPB and UPB leading to a complex dynamics.

2) The assumed Rabi splittings to me seem pretty high. As far as I know the maximum reported Rabi splittings are about 300 meV, but for ensembles of molecules, while for single molecules the Rabi splitting would be well below 100 meV, where according to Fig. 4 the oscillations can hardly be seen, but rather a cavity ring down shows up.

3) It is totally unclear to me how to measure emission with 1 fs time-resolution? The resolution of photodetectors is typically much worse. Also sampling techniques such as luminescence up-conversion typically have clearly higher time-resolution.

4) Even if all the conditions above can be fulfilled, it is not obvious to me that the method also allows one to map the dynamics not only of simple molecules, but also of more complex ones.

In summary, at this point I cannot recommend publication of the manuscript in Nature Communications.

Referee 1

1. The exact definition of the radiative emission E_R is not given. From the plot labels in the figures it appears to be adimensional, while I would have guessed it is the number of photon emitted per unit of time. Please clarify (in the main text).

The time-dependent radiative emission is just the cavity population times the radiative decay rate constant of the cavity, i.e., $E_R = \gamma_{c,r}(\alpha + \alpha)$. As $\gamma_{c,r}$ is a heavily system-dependent quantity and corresponds to a constant prefactor, we set it to unity in the plots for simplicity. We have now clarified this in the text:

“The instantaneous radiative emission rate from the cavity is given by $E_R = \gamma_{c,r}(\alpha + \alpha)$, where $\gamma_{c,r}$ is the radiative decay rate of the cavity excitations. As it corresponds to a constant (system-dependent) factor, we set it to unity in the figures shown in the following. Estimates of the achievable photon yields in realistic systems are given in the discussion section.”

2. There are two points that require a deeper discussion than what it is now in the manuscript:

2.a On p. 5 it is stated that the number of photons produced is low because a single excitation is imparted to the molecule. I think it should also be discussed that not all the photons lost from the cavity can really be detected, since a fraction of them will be irreversibly absorbed by the cavity material (this is certainly true for plasmonic nanocavities). Looking at the weak-coupling limit, the revealed emission from the nanocavity is essentially molecular fluorescence (although occurring well before full relaxation of the excited state), that may be quenched (i.e., most of the photons dissipated inside the cavity material) in this setup. The authors have certainly the tools to provide a quantitative estimate of this effect based on electromagnetic calculations on realistic plasmonic nanocavities, but a qualitative discussion of the point is also enough.

We thank the referee for raising this important point. It is of course true that not all photons lost from the cavity will be emitted as free space photons, and in our setup, the radiative decay channel needs to be significant to allow the detection of the cavity decay. At the same time, the plasmonic nanocavity must optimize the coupling strength to the molecule in order to achieve strong coupling with a single or few emitters. Fortunately, the plasmonic nanocavity that has shown the largest single-molecule coupling strength up to now, the nanoparticle-on-mirror (NPoM), also gives radiative quantum yields ($\eta = \gamma_{c,r}/\gamma_c$) as large as 0.5 for gap sizes of 4 nm (Kongsuwan *et al*, ACS Photonics 5, 186 (2018)). Similar and even larger quantum yields up to 0.6 have been achieved with nanocube antennas (Nature Photonics 8, 835 (2014)), although to the best of our knowledge strong coupling has not been achieved in such cavities yet. To clarify this point, we have added the following sentence in the manuscript:

“Finally, we mention that while the cavity decay rate γ_c in a plasmonic cavity is typi-

cally large and leads to few-femtosecond lifetimes as required for the discussed approach, this rate is often dominated by nonradiative contributions that do not lead to far-field emission. However, fortunately exactly the same plasmonic nanocavity architectures that provide the current largest coupling strengths, such as nanoparticle-on-mirror geometries, also provide a significant radiative quantum yield of close to 50 percent [51-53].”

2.b Excitation of the molecule is described by a classical pulse whose central frequency is resonant with a hybrid light-molecule state, i.e., close to the frequency of the cavity mode. Wouldn't this deform the shape of the pulse? Isn't a description in which the external light source is actually injecting photons in the cavity a more faithful representation of the initial excitation conditions? Please comment

We fully agree that the more realistic description is given by the external light pulse injecting photons into the cavity. This is in fact exactly the approach we use: In our model the external light source drives the cavity mode, see Eq. 1, which leads to an effective distortion of the field felt by the molecule. Due to the short lifetime of the cavity modes we study (on the order of the pulse duration), this distortion is however not large. We have clarified this in the main text: “We note that since the cavity mode is driven by the external field, the effective pulse felt by the molecule (in particular in the weak-coupling limit) is slightly distorted and not just given by $\mathbf{E}(\tau)$.”

3. Details of the TDVMPS calculations should be provided in full to assure reproducibility (and understanding of what is really included), perhaps as SI. It would also be useful to have systematic insights in the interplay between coupling strengths and dephasing times on the expected signal.

In order to fully address this point, we have now added a Supplementary Information, as requested by the referee. This includes a separate section with strongly expanded details on the TDVMPS calculations, as well giving several more references to the relevant literature.

Regarding the interplay between coupling strengths and dephasing times, it turns out that the chain mapping procedure employed in the TDVMPS approach can also provide direct insight on the importance of additional dephasing due to the presence of vibrational modes. Indeed, these properties are strongly molecule-dependent, and by comparing successive hopping strengths in the chain one may get an idea of the importance of vibrational dephasing. We have added the following sentences in the Supplementary Information:

“Interestingly, the chain mapping procedure, which is typically seen as a mostly numerical tool, can also provide direct insight into the dephasing times due to vibrational motion. By comparing the hopping strengths between the reaction coordinate mode and the exciton and the reaction coordinate and the second mode in the chain, one may estimate the importance of additional dephasing due to the presence of more than one vibrational mode. For instance, if the first hopping is much larger than the second one,

as it happens in anthracene, a Holstein-type model that takes into account only a single vibrational mode can be expected to capture the relevant physics at short times.”

4. The abstract makes reference to a class of pump-probe experiments (those where the probe dissociate or ionize the molecules) that is certainly important but not the only one, nor the one that “competes” more closely with the experiment suggested here. Pump-probe experiments where the probe pulse is a weak one, providing transient absorption data, is much more common for molecules in condensed phases. I would rather (or additionally) compare the benefits of the present idea with this kind of experiments.

We agree with the referee that transient absorption spectroscopy also provides a powerful pump-probe setup that is indeed somewhat closer in spirit to our proposed setup. We have added the following sentences to the introduction to provide the necessary context: “Another powerful approach is given by transient absorption spectroscopy, where the change of the absorption spectrum of a probe pulse is monitored as a function of time delay after a pump pulse. While this can provide significant insight about molecular dynamics [38], the interpretation of the spectra is nontrivial due to the competition between several distinct effects (such as ground-state bleach, stimulated emission, and excited-state absorption) in the spectrum [39], such that transient absorption spectroscopy only gives an indirect fingerprint of the molecular dynamics.”

5. The PESs used here are harmonic. Nevertheless, a lot of interesting photochemical processes as photoisomerization or photoreactions would occur on PESs with more complex shapes. For the benefit of the impact of this work, a discussion of what one would expect to reveal for these more complex shapes would be useful, to avoid giving the impression the idea is limited to very simple PESs.

We thank the referee for this question, and fully agree with this point. This question is very much related to question number 4 of referee 2. For that reason, we reply to both here:

In order to address this in detail, we have investigated several additional molecular potentials that provide even richer dynamics (an anharmonic Morse oscillator as well as a dissociative potential), and have included two new figures as well as the corresponding discussion (the Morse potential in the main text and the dissociative potential in the Supplementary Information). In particular, the new Fig. 6 shows that the polaritonic molecular clock provides a direct mapping of the classical energy-dependent oscillation period of an anharmonic oscillator, and also gives a direct mapping of dissociative wavepacket motion for a model molecule with PES similar to those of methyl iodide, see Fig. S1 in the Supplementary Information and the corresponding discussion.

6. In Figure 3a there is a problem with the colorbar scale, as 5.7 and 5.2 appear between 0.0 and 1.0.

We thank the referee for pointing out this typo, which we have corrected.

Referee 2

1. *The time scales are a bit hazy. In order to observe the calculated oscillations clearly, one needs a time-resolution of a fs or even better. This corresponds to a spectral width above 1 eV. On the other hand, for clarity one would like to monitor, for example, only the lower polariton, in particular as the presence of a broad vibrational mode spectrum blurs the oscillatory decay. But how is this possible, as the Rabi-splitting is well below (see also point 2) the spectral window that has to be monitored for obtaining the proper time-resolution? When selecting only the LPB, on the other hand, the time resolution would be typically above 10 fs, smearing out the described dynamics. Furthermore, doesn't the excitation laser pulse duration, assumed to be a few fs, wash out some of the effects? If using a 1 fs-pulse (hard to achieve at 3.5 eV), one would excite both LPB and UPB leading to a complex dynamics.*

We thank the referee for pointing out that these relations should be clarified. Importantly, the excited states that are accessed in the system are fully selected by the initial driving laser pulse, and are in particular determined by its central frequency and duration. There is indeed a tradeoff between duration (which should be short to ensure that a “nice” coherent wave packet is launched) and spectral bandwidth (which should be narrow enough to selectively excite a given manifold). We note that these effects are fully included in our simulation as we treat the driving pulse explicitly in our simulation. In particular, for the pulse duration we choose (spectral bandwidth $\sigma_L = 0.1$ eV, corresponding to a FWHM duration of ≈ 11 fs), it is indeed possible to selectively excite either a lower-polariton or upper-polariton wavepacket (for large enough Rabi splitting), while still starting a well-defined nuclear wavepacket that is not too smeared out in space. This is seen, e.g., in Fig. 2 and Fig. 4, where a significant change of the signal as a function of central frequency is observed. For a too short pulse, the spectrum would become almost independent of the central laser frequency.

After the initial excitation, only the energy-integrated total photon emission rate needs to be monitored and thus no further energy resolution is necessary. This also implies that no trade-off between energy and time resolution in the detection setup is required.

We add to the manuscript a sentence that states the duration of the pulse:

“For $\sigma_L = 0.1$ eV, the duration of the pulse is ≈ 11 fs.”

2. *The assumed Rabi splittings to me seem pretty high. As far as I know the maximum reported Rabi splittings are about 300 meV, but for ensembles of molecules, while for single molecules the Rabi splitting would be well below 100 meV, where according to Fig. 4 the oscillations can hardly be seen, but rather a cavity ring down shows up.*

We thank the referee for raising this point. While for (large) molecular ensembles, Rabi splittings above 1 eV have been observed (see, e.g., Kéna-Cohen et al., Adv. Opt. Mater. 1, 827 (2013)), the highest observed value (to the best of our knowledge) for the case of

single-molecule strong coupling is indeed ≈ 80 meV (Nature 535, 127 (2016)). However, with just 4 molecules a Rabi splitting of ≈ 200 meV is achieved.

Motivated by this, we have now performed explicit calculations showing that our approach also works to observe the (collective) coherent wavepacket dynamics in small ensembles of molecules. In particular, we have performed calculations for the case of collective strong coupling with multiple molecules (up to 4), both with a single vibrational mode and with the full vibrational spectral density using the TDVMPS approach, see Fig. 5 of the main text and Figs. S2 and S3 of the Supplementary Information and the corresponding discussion. These results show that the polaritonic molecular clock can also be used to observe coherent many-molecule wave-packet motion within several molecules at the same time. At the same time, the possibility to use collective strong coupling severely reduces the experimental challenges involved in achieving strong coupling. In that sense, we think that our proposal is within experimental reach.

3. It is totally unclear to me how to measure emission with 1 fs time-resolution? The resolution of photodetectors is typically much worse. Also sampling techniques such as luminescence up-conversion typically have clearly higher time-resolution.

In order to clarify this point, we have updated the paragraph discussing this (at the end of the introduction section) to state explicitly that the required time resolution is on the few-femtosecond scale (as opposed to 1 fs). Such resolution is available with several techniques from the field of ultrafast laser science. In the previous version of the manuscript, we already made reference to SPIDER, FROG and d-scan, and we have now additionally added a reference to the experimentally more straightforward technique of intensity cross-correlation.

4. Even if all the conditions above can be fulfilled, it is not obvious to me that the method also allows one to map the dynamics not only of simple molecules, but also of more complex ones.

We thank the referee for also raising this question that helped us to improve the quality of the manuscript and its generality. As it is closely related to question number 5 of referee 1, we have addressed both points together (see above).

Reviewers' Comments:

Reviewer #1:

Remarks to the Author:

The authors addressed properly my comments. There is still an ambiguity related to the TDVMPS calculations that may be corrected at proof stage. In the SI (and similarly in the main text) it is stated that:

"We thus compare these model calculations with calculations performed using time-dependent variational matrix product states (TDVMPS) approach [2, 3] in which the full phononic spectral density, describing all vibrational modes of the molecule and surroundings, is taken into account."

I can see that the vibrational parameters (frequencies and electron-phonon couplings) for all the vibrational modes of the *molecule* are obtained by TDDFT calculations as stated in the main text (reminding this in the SI would be useful), but what about the modes of *surroundings*? If they are neglected, please state so; if they are accounted at some level please provide the used parameters.

Reviewer #2:

Remarks to the Author:

With interest I have read the resubmission by the authors. One has to admit that they have taken all the points raised by the Referees very and answered as good as one can possibly expect, in a few cases substantiated even with new model calculations.

I am not totally convinced that one would want to detect the emitted light without any spectral filtering as one typically has to fight with "stray light" photons.

I am also still not totally sure that a few ps time resolution is sufficient as claimed by the authors. As one can see from Figure 2 and 3, the separation between two adjacent maxima is 20 fs, i.e. the separation between maximum and minimum is about 10 fs. Every experimentalist would target then a resolution that is an order of magnitude better.

Thus, I am not totally convinced that this technique will give results superior to pump-probe (which can be also applied in the low intensity limit), but the manuscript is technically correct. That the photon energy will depend on the nuclear configuration is clear but the suggestion to use it to determine the nuclear motion is novel, if this kind of reverse engineering problem can be solved uniquely.

As a result, the question of publication boils down to the fact how important the results will become. Probably this can be assessed only after the potential of the technique is tested in experiment.

In summary, I am still skeptical and thus hesitate about recommending publication, but on the other hand also do not want to block it, if the other referee(s) support(s) publication.

Referee 1

1. *There is still an ambiguity related to the TDVMPS calculations that may be corrected at proof stage. In the SI (and similarly in the main text) it is stated that: "We thus compare these model calculations with calculations performed using time-dependent variational matrix product states (TDVMPS) approach [2, 3] in which the full phononic spectral density, describing all vibrational modes of the molecule and surroundings, is taken into account."*

*I can see that the vibrational parameters (frequencies and electron-phonon couplings) for all the vibrational modes of the *molecule* are obtained by TDDFT calculations as stated in the main text (reminding this in the SI would be useful), but what about the modes of *surroundings*? If they are neglected, please state so; if they are accounted at some level please provide the used parameters.*

We added in the SI that the vibrational parameters are obtained by TDDFT calculations. Furthermore, we have added a sentence that clarifies how the surrounding vibrational modes are taken into account.

“The decoherence due to the coupling of the vibrational molecular modes with the surrounding bath is taken into account empirically by replacing the discrete peaks in the spectral density by a Lorentzian with 0.3 meV of width. This does not affect our results.”